# Bias of Attentional Oscillations in Individuals with Subthreshold Depression: Evidence from a Pre-Cueing Facial Expression Judgment Task

**DOI:** 10.3390/ijerph192114559

**Published:** 2022-11-06

**Authors:** Wenfeng Wu, Xiaojiaqi Huang, Xin Qi, Yongbiao Lu

**Affiliations:** Psychological School, Guizhou Normal University, Guiyang 550025, China

**Keywords:** attentional oscillation, attentional bias, alpha neural rhythms, subthreshold depression

## Abstract

**Background**: Study results regarding attentional bias in depressed individuals are inconsistent. Recent studies have found that attention is a discrete process, alternating between periods of either enhanced or diminished attention sensitivity. Whether a visual target can be detected depends on when it occurs relative to these oscillation rhythms. We infer that the inconsistency of attentional bias may be related to the abnormality of attentional oscillations in depressed individuals. **Methods**: A pre-cueing attentional task was used. We set 48 levels of stimulus onset asynchrony (SOA) between cues and targets and measured the response time (RT) of participants, as well as their EEG signals. **Results**: The RTs showed patterns of behavioral oscillations. Repeated-measure ANOVA indicated that subthreshold depressed participants had significantly higher RTs for negative expressions than for neutral but significantly lower RTs for positive than for neutral. The frequency analysis indicated that the RT oscillational frequency of subthreshold depressed participants to negative/positive expressions was different from that to neutral. The EEG time–frequency analysis showed that when faced with negative expressions, the intensity of the neural alpha oscillatory power of subthreshold depressed participants was significantly lower than that of normal controls. When faced with positive expressions, the intensity of neural alpha oscillatory power was significantly higher than that of normal controls. **Conclusion**: Compared to normal persons, subthreshold depressed individuals may have biases in both the amplitude and frequency of attentional oscillations. These attentional biases correspond to the intensity of their neural alpha wave rhythms.

## 1. Introduction

According to cognitive models of depression, attentional biases are assumed to play a critical role in the etiopathogenesis of depression [1]. Beck thought that internal mental schemas affect how depressed persons perceive themselves and the world around them. Depressed individuals have mood-congruent schemas that are characterized by themes of loss, worthlessness, and rejection; these schemas lead these individuals to have negative perceptions of themselves, the world, and the future and cause them to display negative information-processing biases [2]. A considerable amount of research has revealed that not only depressed patients but also persons at risk for depression prefer to pay more attention to negative stimuli [3]. However, whether depressed individuals are characterized by negative biases in attention has still been a topic of considerable debate [4]. Additionally, some researchers did not find negative attentional bias in depressed individuals in their studies [5,6].

In the early days, researchers mainly focused on the study of depressive attentional bias to negative emotional stimuli based on Beck’s theory. However, in subsequent studies, it was found that in contrast to negative emotional stimuli, when positive emotional stimuli were presented, the results of depressive attentional bias were more consistent. Suslow, Junghanns and Arolt [7] found that depressed participants showed no performance differences in their detection of negative emotional faces compared to normal participants, while they were significantly slower in responding to positive emotional faces than normal participants. Another meta-analysis of attentional bias studies using eye tracking also found that depressed individuals were characterized by reduced orientation to positive stimuli [8]. To explain the positive bias, the “reward devaluation” hypothesis claims that depressed individuals automatically avoid positive information [9].

However, research has also found neither negative bias nor positive bias in depressed individuals’ attention [6]. Evidence of attention bias in depression has been equivocal [4]. Historically, most attentional bias studies concerning depression have generally used the emotional Stroop task or the dot-probe task, and the latter has become the most-used paradigm [4,10]. The dot-probe task requires participants to focus first on a pair of images or words that appear in negative and neutral pairs on either side of the screen; after these images or words disappear, there follows a period of time, the stimulus onset asynchrony (SOA), and then a dot appears on the location where that negative or neutral image (word) disappeared. Participants are asked to respond to the dot as quickly as possible. Attention bias scores are calculated by comparing response times to the dots that appear in the same location as neutral cues with those to the dots that appear in the same location as emotional cues. There is substantial heterogeneity in the results, and mixed findings have been reported with regards to attention bias [11].

Classic studies of spatial attention have largely ignored temporal dynamics, thus assuming that associated neural and behavioral effects are continuous over time and persist for the duration of attentional deployment [12,13]. Recent studies, however, have found that there are oscillations in attention [12,14,15]. Attention is not a continuous process but rather a discrete one, alternating between periods of either enhanced or diminished attention sensitivity. Whether a visual target can be detected depends not only on its stimulus properties but also on when it occurs relative to these oscillation rhythms [12]. By setting multiple levels of SOA, researchers have found that the performance of the participants’ attention changes rhythmically with the SOA, performing the behavioral oscillation phenomenon of attention [16,17,18]. A meta-analysis found that the temporal dynamics of the biased attentional processing of emotional information were significantly more elevated among remitted depressed patients than among healthy controls [19]. According to these findings regarding attentional oscillations, we speculate that the inconsistency of attentional bias studies may be related to the attentional oscillations of depressed individuals. The attentional oscillations of depressed individuals may be different from those of normal individuals, causing inconsistent study results.

Research has demonstrated that attention biases manifest not only in people with major depression but also in individuals at risk for depression [3,20]. Among the risk factors for major depression, the presence of subthreshold depression (SD) is one of the most central factors. SD refers to clinically relevant depressive symptoms that do not meet the criteria for major depression [21]. Some researchers even suggest that SD is a mild form of major depression [22]. Given that SD is a significant precursor to and a major risk factor for major depressive disorder and that subthreshold depressive individuals have attentional bias [22,23], then SD is also closely related to individuals’ negative results, such as an increased burden of disease, impaired functioning, and suicide risk [24]. We decided to select subthreshold depressive individuals as our experimental participants. Based on previous research regarding depressive attention bias, we submitted three hypotheses: (I) The attentional oscillations of subthreshold depressive individuals significantly differ from those of normal persons. (II) For a negative emotional stimulus, the range of attentional oscillation amplitude of subthreshold depressed participants is significantly lower than that for a neutral stimulus. That is, the overall mean RT to a negative emotional stimulus is remarkably lower than that to a neutral stimulus. (III) For a positive emotional stimulus, the range of the attentional oscillation amplitude of subthreshold depressed participants is significantly higher than that for a neutral stimulus.

Studies have found that attentional behavioral oscillations are related to periodic rhythms of neural activity; they have the same rhythmic components and involve similar mental processes [25,26,27]. The response of neurons does not merely depend on external inputs; in fact, a repeated presentation of the same stimulus gives rise to highly variable responses at the behavioral level as well as at the neural level [28]. Some studies have demonstrated that the time–frequency pattern of brain oscillations may represent a key feature that shapes both neural and behavioral responses by its ongoing activity [29,30,31]. Research has discovered that the primary visual cortex modulates attention oscillation [32]. Du et al. [33] recently found that there are visual cortical excitability alterations related to psychopathological symptoms in major depressive disorder. Based on the related findings, we propose hypothesis (IV): Corresponding to attentional behavior oscillation, when paying attention to a negative or positive emotional expression, the rhythms of neural activity in the visual cortex of subthreshold depressive individuals are different from those of normal persons.

## 2. Materials and Methods

### 2.1. Participants

Setting effect size = 0.25, α = 0.05, 1 − *β* = 0.95, used G-power to calculate that the total number of required participants was 33. In Experiment I, 46 college students were recruited, of whom 24 were placed in the SD group and 22 were placed in the control group. All participants had normal or corrected-to-normal vision. Due to the failure of some participants to complete the experiment and to equipment faults, 6 participants were excluded, including 4 in the SD group and 2 in the control group. Among the valid participants in the SD group, the mean age was 20.30 ± 2.32, including 9 males and 11 females; in the control group, the mean age was 19.80 ± 1.36, including 6 males and 14 females. In experiment II, the SD group had 23 students, and the control group had 25 students. Because of incomplete experiments and the error rate over 20%, 3 students in the SD group and 5 students in the control group were excluded.

### 2.2. Selection of Participants

Referring to previous research [34,35], we first used the method of cluster sampling, during which college students were asked to fill in the Center for Epidemiologic Studies-Depression scale (CES-D) [36] and Self-rating Depression Scale (SDS) [37]. Second, students whose total CES-D score was over 20 and whose standard SDS score was over 50 were invited to our lab and received a one-on-one screening interview based on the depressive diagnosis criteria of the DSM-IV to exclude participants who met the following criteria: (1) have met the diagnosis criteria of major depression; and (2) have experienced major negative events within 3 months. Interviews were conducted by a single interviewer who had earned a national psychological consultant level 3 certificate (Issued by the Ministry of Human Resources and Social Security of the People’s Republic of China) and was trained in DSM-IV administration and scoring, and who was blind to participants’ measurement results of depressive symptoms.

### 2.3. Materials

Forty neutral facial pictures and 40 negative emotional facial pictures were selected for Experiment I, and 40 neutral facial pictures (same as those in Experiment I) and 40 positive emotional facial pictures were selected for experiment II from the Chinese Facial Affective Picture System (CFAPS) [38]. The valence of negative emotional expressions was significantly lower than that of neutral expressions (t = −11.4, *p* < 0.001), and the valence of positive emotional expressions was remarkably higher than that of neutral expressions (t = 9.57, *p* < 0.001).

### 2.4. Experimental Designs and Procedure

A total of 2 (Group: SD and control) × 2 (Cue: valid and invalid) × 2 (Valence: negative/positive and neutral) mixed designs were used. The group was used as a between-participants variable, while the cue and valence were used as within-participants variables. In order to explore the attentional oscillations, referring to the study of Song et al. [18], the cue–target experimental paradigm was applied, the SOA was set to 48 levels, ranging from 200 ms to 1100 ms and increasing by 20 ms for each level up. The RT of participants and their EEG event-related potentials were used as independent variables. The experimental procedure is displayed in Figure 1. At the beginning of every trial, a blank screen was displayed randomly for 1–1.5 s, then a central fixation cross and two white boxes appeared in the center of the screen (random for 0.8–1.2 s), after which a white grating appeared next to one of the two boxes as the peripheral clues (with a 50% probability of randomly appearing on the left or the right sides) for 0.1 s. After the white grating disappeared, following different SOA intervals, the target stimulus randomly appeared in one of the two boxes (with a probability of 50%) for 0.15 s. Finally, the participants were required to press the key within 2 s judging whether the facial picture with or without emotion; if there was no response, the next trial began automatically. The setting of SOA = 0.2 s was 10 times that of other SOA settings to obtain the reset effect of attention oscillation.

In Experiment I, the target stimuli were negative and neutral expressions. In experiment II, the target stimuli were positive and neutral expressions. In both Experiments I and II, the probability of the appearance of negative/positive expressions and neutral expressions was 50%, respectively. Before the experiment, the participant had been asked to complete 10 exercises, and their rate of correct responses was required to be over 80% to start the formal experiment.

All participants gave their informed consent and received a subject fee of RMB 50 after the experiment was completed. The total time for data acquisition was approximately one and half hours. Following recommendations in research transparency and replicability, the MATLAB versions of the experimental programs, as well as all the stimuli, are available via the following review link: https://osf.io/f2t5v/files/osfstorage?view_only=ec3962b42ef645b1ae97088e72cda11a (accessed on 6 June 2022)

### 2.5. EEG Data Recording and Analysis

#### 2.5.1. EEG Data Recording

EEG were measured using an electrode cap (Neuroscan, Herndon, VA, USA) with 64 Ag/AgCl scalp sites according to the International 10/20 system. The EEG signal was filtered with an analog filter range of 0.05–100 Hz and then saved at a sampling rate of 1000. In addition to the 64-leads EEG channels, bipolar 2-leads were used to monitor eye electric potentials. All electrode impedances were kept below 5 kΩ during the experiment.

#### 2.5.2. Preparation of EEG Data

Using the plug-in EEGLAB in MATLAB software to preprocess EEG data, we used the whole-brain average for offline re-referencing. EEGLAB’s own filter was applied to filter the signals collected by all EEG electrode points (low pass was 0.01 Hz, high pass was 30 Hz). The timepoint when the face target stimulus appeared was set as the zero point, and the time period from 200 ms before to 600 ms after the appearance of the face target was taken as the time window for making epochs. After that, ICA was used to exclude the influence of eye movement on EEG data. Finally, the trials with potential amplitudes over ±80 µV were rejected.

#### 2.5.3. EEG Time–Frequency Analysis

Using fast Fourier transform (FFT) to decompose the spectra of raw EEG epoch data, the decomposed frequencies were found to be 1 HZ to 30 HZ, and the step was found to be 1 HZ. After that, considering that the interval time of SOA was small, which more than likely directly caused the neural oscillation at the occipital lobe area, we selected the occipital lobe area (left occipital lobe: PO3, PO5, PO7; right occipital lobe: PO4, PO6, PO8) on which to draw the time–frequency diagram of alpha oscillation by open window.

## 3. Results

### 3.1. Results of RT

For every group participant, we first calculated the RTs of each SOA under valid and invalid cue conditions and normalized the RTs. Next, we ranked the RTs of participants by SOA under valid and invalid cues to determine the variation in detection ability over time in the SD group and in the normal control group (RT-SOA signal); see Figure 2.

The graphs show that the data results illustrate a clear oscillation pattern. Figure 2A,C show the RT oscillation diagram of the participants under the four conditions of invalid neutral, valid neutral, invalid negative, and valid negative after detrending the data of the control group. Figure 2B,D show the RT oscillation diagram of the participants under the same four conditions after detrending the data of the SD group. In Figure 2A,C, the RT oscillation amplitudes for the control group are basically the same under both neutral and negative/positive conditions. In Figure 2B, the amplitude values of the SD group participants’ RT oscillations under negative conditions were obviously lower than those under neutral conditions. However, in Figure 2D, the amplitude values of SD participants’ RT oscillations under positive conditions were obviously higher than those under neutral conditions.

To further verify whether there were significant differences in the RT under different conditions, we set RT as the dependent variable and excluded the RT values that exceeded ±3 standard deviations (the exclusion rate was 3.8%). Repeated-measures ANOVAs of 2 (group: subthreshold depression and control) × 2 (cues: valid and invalid) × 2 (emotion valence: negative/positive and neutral) showed that in Experiment I, the main effect of group was not significant (*F*(1, 38) = 0.022, *p* > 0.05), and the main effects of emotion valence and cues were significant (*F*(1, 38) = 90.52, *p* < 0.001, $\eta_{p}^{2}$ = 0.704; *F*(1, 38) =20.42, *p* < 0.001, $\eta_{p}^{2}$ = 0.350). Among the interactions, only the interaction between emotion valence and group was significant (*F*(1, 38) = 84.40, *p* < 0.001, $\eta_{p}^{2}$ = 0.690). Further simple effect analysis revealed that in the SD group, the RT to negative facial expressions was significantly faster than that of control group (*F*(1, 38) = 4.11, *p* < 0.05, $\eta_{p}^{2}$ = 0.098), and the RT to negative facial expressions was significantly faster than that to neutral facial expressions (*F*(1, 38) = 174.86, *p* < 0.001, $\eta_{p}^{2}$ = 0.821). The RT in the control group to negative and neutral expressions was not remarkably different (*F*(1, 38) = 0.054, *p* > 0.05).

In experiment II, the main effect of group was not significant (*F*(1, 38) = 1.75, *p* > 0.05); the main effects of emotional valence and cues were significant (*F*(1, 38) = 26.35, *p* < 0.001, $\eta_{p}^{2}$ = 0.409; *F*(1, 38) = 6.51, *p* = 0.015, $\eta_{p}^{2}$ = 0.146); among the interactions, only the interaction between emotional valence and group was significant (*F*(1, 38) = 65.86, *p* < 0.001, $\eta_{p}^{2}$ = 0.634). A simple effect analysis indicated that the RT in the SD group to positive facial expressions was significantly slower than that of the control group (*F*(1, 38) = 6.234, *p* = 0.017, $\eta_{p}^{2}$ = 0.141); the RT to positive facial expressions was significantly slower than that to neutral facial expressions in the SD group (*F*(1, 38) = 87.758, *p* < 0.001, $\eta_{p}^{2}$ = 0.698). The RT of participants in the control group to positive and neutral expressions was not remarkably different (*F*(1, 38) = 0.042, *p* > 0.05). For intuitive results, please see Figure 3.

To determine whether there was a difference in the RT oscillation frequency under different conditions, using fast Fourier transformation on the data in MATLAB, we transformed the time domain in the behavioral oscillation to the frequency domain and obtained the frequency domain diagram (Figure 4). Referring to the method of Landau and Fries [17], we obtained the frequency range of the peak amplitude. Experiment I showed that for the control group, under neutral emotion valence, the oscillation mode of RT displayed frequencies of 3–7 Hz (the horizontal solid and dotted blue lines represent the valid cues at 6.25–7.03 Hz and the invalid cues at 3.91–4.69 Hz, respectively); under negative valence, the oscillation mode of RT displayed frequencies of 3–6 Hz (the horizontal solid and dotted red lines represent the valid cues at 4.69–5.47 Hz and the invalid cues at 3.31–3.90 Hz, respectively). For SD group, under neutral valence, the oscillation mode of RT displayed frequencies of 6–9 Hz (the horizontal solid and dotted blue lines represent the valid cues at 6.25–7.03 Hz and the invalid cues at 8.59–9.38 Hz, respectively); under negative valence, the oscillation mode of RT displayed frequencies of 4–7 Hz (the horizontal solid and dotted red lines represent the valid cues at 4.69–5.47 Hz and the invalid cues at 6.25–7.03 Hz, respectively). These results indicate that for the control group, regardless of whether neutral or negative stimuli were present, the frequencies of attention oscillation were basically consistent; for the subthreshold depressive participants, however, the frequencies of attentional oscillation were obviously different.

Experiment II showed that for the control group, under neutral valence, the oscillation mode of RT displayed frequencies of 4–6 Hz (the horizontal solid and dotted blue lines represent the valid cues at 4.49–5.47 Hz and the invalid cues at 5.47–6.25 Hz, respectively); under positive valence, the oscillation mode of RT displayed frequencies of 3–8 Hz (the horizontal solid and dotted red lines represent the valid cues at 3.91–4.67 Hz and the invalid cues at 7.03–7.81 Hz, respectively). For the SD group, under neutral valence, the oscillation mode of RT displayed frequencies of 1–4 Hz (the horizontal solid and dotted blue lines represent the valid cues at 1.56–2.34 Hz and the invalid cues at 3.13–3.91 Hz, respectively); under positive valence, the oscillation mode of RT displayed frequencies of 3–7 Hz (the horizontal solid and dotted red lines represent the valid cues at 3.13–3.91 Hz and the invalid cues at 6.25–7.03 Hz, respectively). These results demonstrate that for the control group, regardless of whether they were faced with neutral or positive emotional stimuli, the frequencies of attention oscillation were basically consistent; however, for the SD participants, the frequencies of attentional oscillation were obviously different.

### 3.2. Results of EEG

Time-frequency analysis showed that in Experiment I, after the target stimulus appeared at 0.15–0.2 s, the alpha neural oscillation showed a relatively large energy increase (see Figure 5). Repeated-measures ANOVA indicated that the main effect of cues and emotional valence was significant (*F*(1, 38) = 17.697, *p* > 0.001, $\eta_{p}^{2}$ = 0.318; *F*(1, 38) = 4.11, *p* = 0.050, $\eta_{p}^{2}$ = 0.098). The interaction between group and emotional valence was marginally significant (*F*(1, 38) = 3.59, *p* = 0.066, $\eta_{p}^{2}$ = 0.086). Simple effect analysis indicated, for SD participants, when the emotional valence condition was negative, the energy value of the alpha neural oscillation was significantly lower than that of the control participants (*F*(1, 38) = 11.64, *p* = 0.002, $\eta_{p}^{2}$ = 0.234) and significantly lower than that when the emotional valence was neutral (*F*(1, 38) = 7.69, *p* = 0.009, $\eta_{p}^{2}$ = 0.168). The other effects were not significant (all *p* > 0.05).

In experiment II, the alpha neural oscillation also showed a relatively large energy increase after the target stimulus appeared at 0.15–0.2 s (see Figure 5). Repeated-measure ANOVA indicated that the main effect of cues was significant (*F*(1, 38) = 38.84, *p* < 0.001, $\eta_{p}^{2}$ = 0.505). The main effect of emotional valence was marginally significant (*F*(1, 38) = 3.93, *p* = 0.055, $\eta_{p}^{2}$ = 0.094). The interaction of group and emotional valence was also marginally significant (*F*(1, 38) = 4.051, *p* = 0.051, $\eta_{p}^{2}$ = 0.096). Simple effect analysis indicated, for SD subjects, under the condition of positive valence, the energy value of the alpha neural oscillation was significantly higher than that of control participants (*F*(1, 38) = 16.134, *p* < 0.001, $\eta_{p}^{2}$ = 0.298) and significantly higher than that when the emotion valence was neutral (*F*(1, 38) = 7.980, *p* = 0.007, $\eta_{p}^{2}$ = 0.174). The other effects were not significant (all *p* > 0.05).

## 4. Discussion

This study showed that there were differences in behavioral attentional oscillation patterns between SD participants and normal control participants. The specific performance is described as follows. For the SD participants, the attentional oscillation frequency under the condition of negative and positive stimuli was significantly different from that under the condition of neutral stimuli. However, for the control group participants, the attentional frequency was basically consistent under negative and neutral stimuli or under positive and neutral stimuli. The study also found that participants with SD showed negative emotional attention bias. Compared to the normal group, the location of the RT oscillation amplitude range of the SD group was lower, and its overall mean RT was significantly less than that of the normal control group. In addition, the study found that SD participants demonstrated a decrease in positive emotional attention. Compared to the normal control group, the location of the RT oscillation amplitude range of the SD group was higher, and its overall mean RT was significantly greater than that of the normal group. With respect to neural oscillation, the SD participants also differed from the normal participants.

### 4.1. Differences in Behavioral Oscillations between SD Individuals and Normal Persons

Many researchers have explored the theme of depressive individuals’ attentional bias [4]. Based on the theory and the results of previous studies, three different views regarding the attentional bias of depression have been proposed: (1) depressed individuals show an increased negative reactivity called negative potentiation; (2) they show a reduced positive reactivity called positive attenuation; and (3) they exhibit a reduced positive and negative reactivity called emotion context insensitivity (ECI) [39,40]. Most studies have used the dot-probe task to assess attentional biases. This approach, however, has been found to have poor psychometric properties [41], which may contribute to the equivocality of the findings. Zvielli et al. [11] proposed that attentional bias may be better understood as a dynamic process expressed in fluctuating, phasic bursts. In this study, by measuring SD participants’ RT to negative and neutral emotional expressions, we found that the RT exhibited oscillating variation with different SOAs, and the RT to negative expression was faster than that to neutral expression, supporting the view of negative potentiation. Zvielli et al. [19] used a novel computational approach to extract a series of bias estimations from trial-to-trial and also found that temporal variability in attentional bias was significantly elevated among remitted depressed patients relative to nondepressed controls. At the same time, the results of our study also support the view of positive attenuation. For the SD participants, by measuring their RT to positive and neutral expressions at different SOAs, we found that the RT to positive emotional expression was slower than that to neutral emotional expression, although there was a fluctuation between the two. Most research on positive attenuation did not explore attentional bias but rather explored the bias of working memory, motivation, explanation, etc., and their results verified the view of positive attenuation in depressed persons [42,43].

In addition, the results of our research also verified the hypothesis that the attentional oscillation frequency of SD individuals differs from that of normal persons. Previous studies have found that abnormal neural oscillations are linked to depression [44,45] and that the periodic rhythms of neural activity that are related to attentional behavioral oscillations have the same rhythmic components and involve similar mental processes [25,26,27]. Few studies have explored the difference in attentional oscillation between depressed individuals and normal individuals, and our study supplies new verification information regarding this issue.

### 4.2. Differences in Neural Oscillations between SD Individuals and Normal Persons

Our study found that SD individuals differ from normal persons in the amplitude and frequency of their attentional behavioral oscillations. A time–frequency analysis of EEGs found that there were also differences in alpha neural oscillations in the occipital lobe region between SD individuals and normal persons. Their performance was as follows: when faced with negative stimuli, the amplitude of the neural concussion alpha waves of SD participants was stronger than that of normal control participants, while when faced with positive expression stimuli, the result was the opposite. Helfrich et al. [26] demonstrated that the neural basis of sustained attention is rhythmic. Du et al. [33] also found that the degree of symptoms in individuals with major depressive disorder is associated with altered visual cortical excitability. The research evidence suggests that alpha-band neural oscillations are the dominant oscillations in the human brain and that they have an inhibitory function [46]. Our research demonstrates that when faced with negative facial emotional expressions, the amplitude of the alpha neural oscillation in the occipital visual cortex of the SD participants, corresponding to the faster RT of attentional oscillation, was significantly lower than that of normal participants, indicating that their inhibition to negative stimuli was weakened. When faced with positive expressions, the amplitude of the alpha neural oscillation in the occipital visual cortex of the SD participants, corresponding to the slower response time of attentional oscillation, was significantly stronger than that of normal participants, indicating that their inhibition to positive stimuli was enhanced.

## 5. Conclusions

The attentional oscillational frequencies of subthreshold depressed individuals are different from a normal person.Compared with normal individuals, subthreshold depressed individuals have a bias of the enhancement to negative stimuli and the weakening to positive stimuli in attentional oscillations.These attentional behavioral biases may correspond to their neural alpha wave rhythms in the occipital lobe region.

## Figures and Tables

**Figure 1 ijerph-19-14559-f001:**
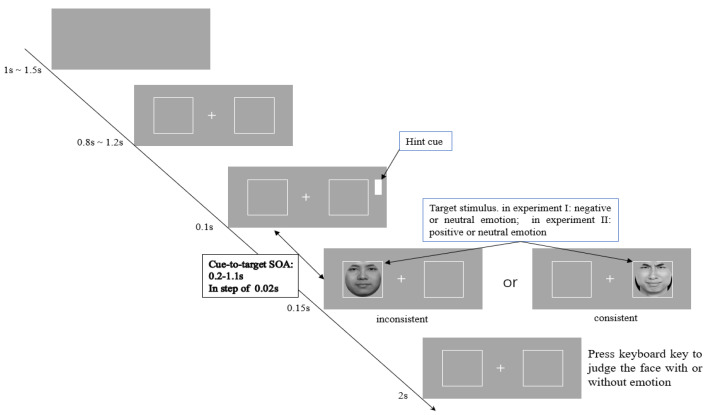
The experimental procedure.

**Figure 2 ijerph-19-14559-f002:**
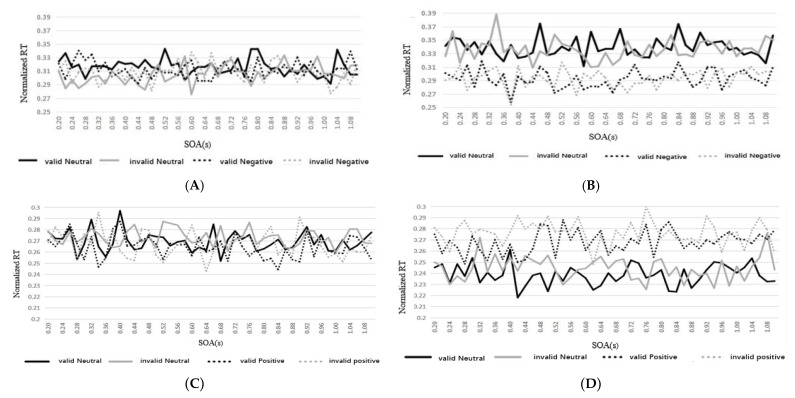
Normalized RT of different SOAs. Experiment I: (**A**) Control group. (**B**) Subthreshold depression group. Experiment II: (**C**) Control group. (**D**) Subthreshold depression group.

**Figure 3 ijerph-19-14559-f003:**
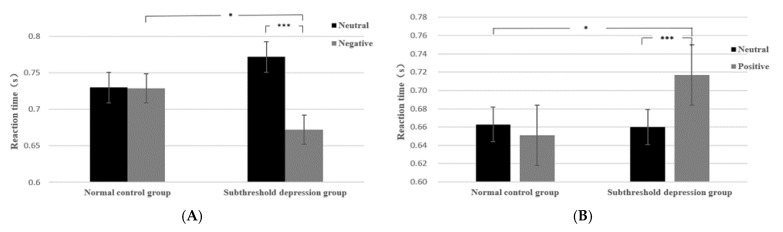
The interaction of valence and group. (**A**) Experiment I. (**B**) Experiment II. * represent *p* < 0.05, *** represent *p* < 0.001.

**Figure 4 ijerph-19-14559-f004:**
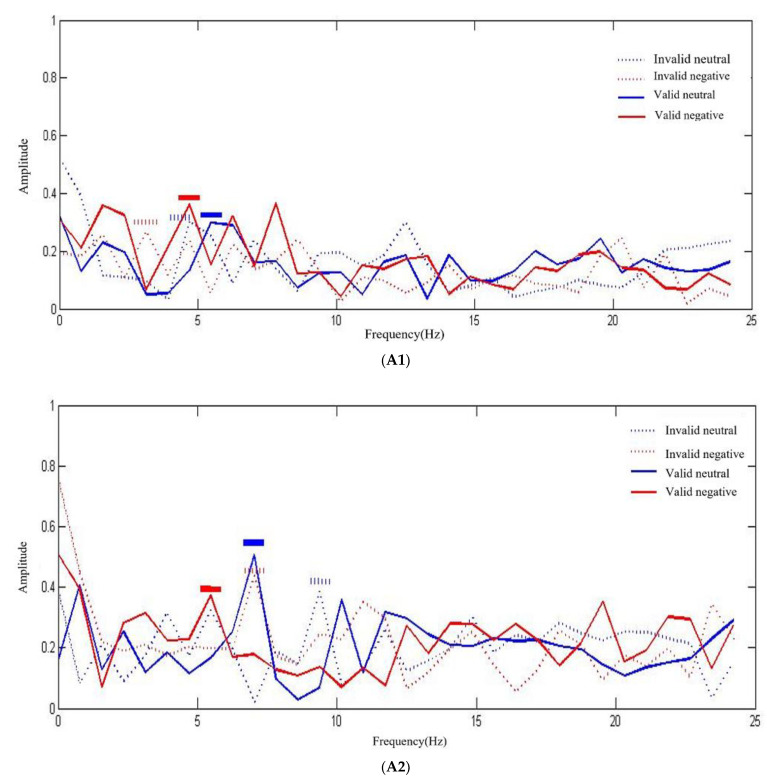
The oscillations of attention. (**A**) Experiment I (**A1** is the control group; **A2** is the subthreshold depression group); (**B**) Experiment II (**B1** is the control group; **B2** is the subthreshold depression group).

**Figure 5 ijerph-19-14559-f005:**
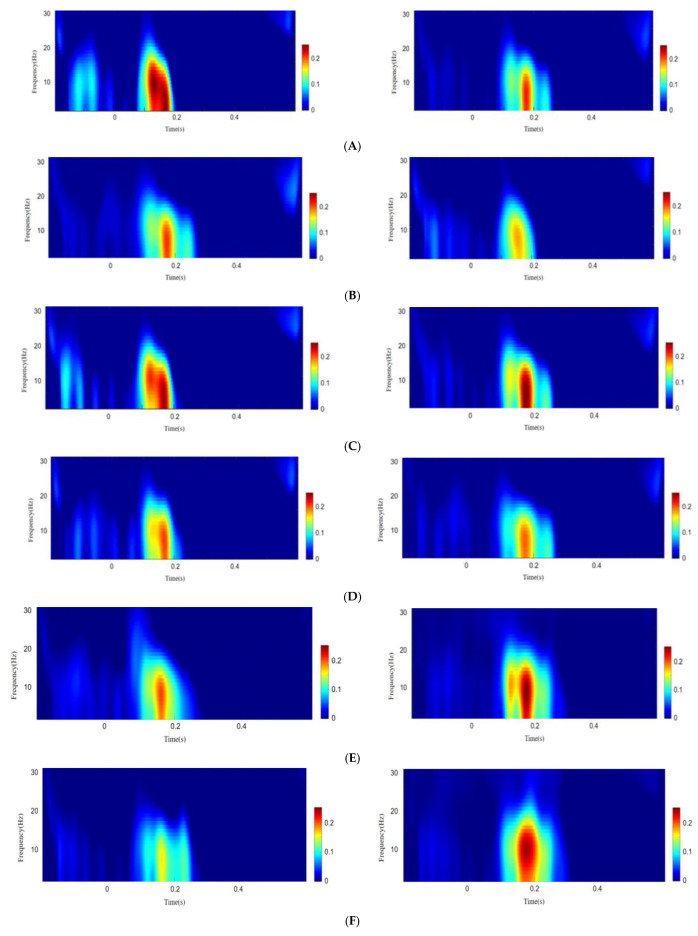
Alpha-band time–frequency diagram. Experiment I: (**A**) Condition of invalid cue and negative emotion (left is control group; right is subthreshold depression group). (**B**) Condition of valid cue and negative emotion (left is control group; right is subthreshold depression group). (**C**) Condition of invalid cue and neutral emotion (left is control group; right is subthreshold depression group). (**D**) Condition of valid cue and neutral emotion (left is the control group; right is the subthreshold depression group). Experiment II: (**E**) Condition of invalid cues and positive emotions (left is the control group; right is the subthreshold depression group). (**F**) Condition of valid cue and positive emotion (left is control group; right is subthreshold depression group). (**G**) Condition of invalid cues and neutral emotions (left is the control group; right is the subthreshold depression group). (**H**) Condition of invalid cues and neutral emotions (left is the control group; right is the subthreshold depression group).

## Data Availability

The data presented in this study are available on request from the corresponding author.

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
