# Peer review of "Bias of Attentional Oscillations in Individuals with Subthreshold Depression: Evidence from a Pre-Cueing Facial Expression Judgment Task"

_ijerph, 2022, doi:10.3390/ijerph192114559_

Round 1

Reviewer 1 Report

Dear authors,

I have read the manuscript entitled "Bias of attentional oscillations in individuals with subthresh-old depression: Evidence from a pre-cueing facial expression judgment task".

This original clinical study, conducted in a young population, demonstrated that attentional biases in depressed individuals cannot be confounded. Statistical analysis based on repeated measures indicated that participants below the depressive threshold had significantly longer response times for the negative facial expression. This study reveals that individuals with subthreshold depression differ from normal individuals in the amplitude and frequency of behavioral attentional oscillations compared to normal individuals.

The paper complies with the requirements of the journal, the results obtained were highlighted by means of figures, giving the reader the possibility to make the reading easier. The bibliographic references used for documentation are in accordance with the chosen topic.

However, I have a few minor questions and suggestions:

1. Reading the manuscript is a bit difficult due to the convoluted manner of expression. Authors should consider this aspect and to clarify the content a little more.

2. In the Materials and Methods section you mentioned using a number of 40 neutral facial images and 40 negative facial images. Participants after viewing each image responded within a certain time. Can the authors argue what the accuracy of this assessment would be? Were there any participants whose vision needed to be corrected with glasses?

3. Based on the obtained results, can the authors argue whether such studies could be used as screening in detecting the population at risk of depression?

4. It is well known that alpha stimulation helps with emotional stabilization, a damaged state in depressed patients or those with subthreshold depression. You stated that alpha waves differ in intensity and frequency depending on the type of visual image. What would happen to alpha waves if the participants viewed positive, happy images, images that would not create an unpleasant state for the participants?

5. Although the Conclusions section is optional, I appreciate you giving us some. However, it would be welcome for readers if the authors would like to expand the number of conclusions and be a little clearer.

6. For some figures, I recommend a larger font size.

7. English must be improved.

Author Response

Thanks for reviewer’s comments! According to your suggestions, we have revised the manuscript one by one as follow:

  1. Reading the manuscript is a bit difficult due to the convoluted manner of expression. Authors should consider this aspect and to clarify the content a little more.

Thanks for your advice!As the non-native English-speaking authors, we are aware that there may be language deficiencies for us, so after the manuscript completed, we ordered American Journal Experts (AJE) company to polish the full text, and the company promised that they edited the manuscript for correct English language, grammar, punctuation, and phrasing. In addition, based on their extensive knowledge of what journals typically require, they also had made some changes to ensure consistency throughout the paper. The verification code: D31A-E96D-0813-F163-7C98.

Based on your suggestion, we will carefully read the manuscript more times and try our best to improve it.

  1. In the Materials and Methods section you mentioned using a number of 40 neutral facial images and 40 negative facial images. Participants after viewing each image responded within a certain time. Can the authors argue what the accuracy of this assessment would be? Were there any participants whose vision needed to be corrected with glasses?

Thanks for your reminder! In the manuscript, we have mentioned “Because the reason of uncompleted the experiment and the error rate over 20%, 3 students in the SD group and 5 students in the control group were excluded.” The 3 students in the SD group and 5 students in the control group were the participants who made mistake selection of neutral facial images and negative/positive facial images over 20%. In addition, based on your opinions, we have added the content of that” All participants had normal or corrected-to-normal vision.” Please see the participants section in the revised manuscript.

  1. Based on the obtained results, can the authors argue whether such studies could be used as screening in detecting the population at risk of depression?

Thanks for your nice idea! That is our pursuing goal, however since we have not found other studies exploring this question, we think maybe it is too early to use our findings as a screen to detect people at risk of depression, which may require more additional research to support these results.

  1. It is well known that alpha stimulation helps with emotional stabilization, a damaged state in depressed patients or those with subthreshold depression. You stated that alpha waves differ in intensity and frequency depending on the type of visual image. What would happen to alpha waves if the participants viewed positive, happy images, images that would not create an unpleasant state for the participants?

In our research, when faced with positive expressions (e.g., happy), the amplitude of the alpha band in the occipital visual cortex of the SD participants was significantly stronger than that of normal participants, indicating that their inhibition to positive stimuli was enhanced.

  1. Although the Conclusions section is optional, I appreciate you giving us some. However, it would be welcome for readers if the authors would like to expand the number of conclusions and be a little clearer.

Thank you for your suggestions! We have revised the discussions section according to your advice. Please see the “Track changes” in this section.

  1. For some figures, I recommend a larger font size.

Thank you for your good suggestions, we have adjusted the font size larger in the figure2 and 3.

  1. English must be improved.

The English is very important for an article, so, we ordered American Journal Experts (AJE) company to polish the full text after finished the manuscript. Anyway, according to your opinion, we have again carefully read the manuscript and tried our best to improve it. For details, please see the part of “Track changes”.

Reviewer 2 Report

The four hypotheses proposed in this work are validated by measuring the Response Time over a group of interest participating in the experimental tests. Figures and plots help to understand the performed experiments and the validation of results. The document shows minor edited mistakes, such as capital letters after a period, but the testing framework is generally understandable and well-described. However, I found some concerns about the proposed work:

1)    I understand that the CES-D tests provide a score to evaluate the depression degree in the applicants. But the authors should include more information about CES-D and SDS scores in terms of who validates (or reviews) the tests of participants, an expert? And which expertise do they have (or do they require)?

2)    Which are the advantages of using an ANOVA analysis regarding other metrics? How does the ANOVA analysis allow authors to generalize the RT behavior for participants? That is, which other metrics could be included?

3)    Similar hypotheses and statements proposed in this work have been validated. How does the proposed experimental test show a notable advantage?

4)    On page 4, the link does not contain the dataset; could the authors provide the EEG signals recorded?

5)    Could the author include what is next in this research? What will be the use of these results?

Author Response

Thanks for reviewer’s comments! According to your suggestions, we have revised the manuscript one by one as follow:

(1) I understand that the CES-D tests provide a score to evaluate the depression degree in the applicants. But the authors should include more information about CES-D and SDS scores in terms of who validates (or reviews) the tests of participants, an expert? And which expertise do they have (or do they require)?

Thanks for your advice! We have added the related content in revised manuscript. Please see the part of “Track changes”

(2) Which are the advantages of using an ANOVA analysis regarding other metrics? How does the ANOVA analysis allow authors to generalize the RT behavior for participants? That is, which other metrics could be included?

   Because our study is a 2 (group: SD and control) × 2 (hint: effective and ineffective) × 2 (negative/positive and neutral) mixed experimental design, which is a multifactor experimental design. referring to most studies, ANOVA analysis is a popular method to deal with this kind of experimental data, for this reason, we chose this method to analysis RT. Of course, our study took the approach of dynamic attention studies, perhaps it is possible to analyze these data through dynamic structural equation model.

(3) Similar hypotheses and statements proposed in this work have been validated. How does the proposed experimental test show a notable advantage?

Thanks for your review! As you said: “Similar hypotheses and statements proposed in this work have been validated”. However, the classic studies of spatial attention have largely ignored temporal dynamics and thus does not fully consider attentional bias results may be affected by attentional oscillations. Referring to the findings of attentional oscillation research, this study adopts the method of presenting stimuli at dynamic intervals to examine the attentional bias of subthreshold depressive individuals in the case of attentional oscillation, which has certain advantages compared with previous studies.  

4)    On page 4, the link does not contain the dataset; could the authors provide the EEG signals recorded?

   Thanks for your advice! We plan to upload our data after carefully edited.

5)    Could the author include what is next in this research? What will be the use of these results?

    Thanks for your questions! First, we plan to conduct the research on individuals with major depression. This may give us more power to infer that abnormal attentional oscillations of depressed individuals may be their biomarker. Second, this study preliminarily revealed that the attentional oscillations of subthreshold depressed individuals were related to their intensity of neural rhythms in the occipital lobe region but did not explore the possible causal relation of them. For further experiments, we plan to use transcranial electrical stimulation or transcranial magnetic stimulation to modulate the neural rhythm of depressed persons and observe whether their attentional oscillation bias will accordingly change to reveal their causal relationship. So that to provide reference for clinical treatment of depression.